# The Challenge to Stabilize, Extract and Analyze Urinary Cell-Free DNA (ucfDNA) during Clinical Routine

**DOI:** 10.3390/diagnostics13243670

**Published:** 2023-12-14

**Authors:** Ivonne Nel, Carolin Münch, Saikal Shamkeeva, Mitja L. Heinemann, Berend Isermann, Bahriye Aktas

**Affiliations:** 1Department of Gynecology, Medical Center, University of Leipzig, 04103 Leipzig, Germany; 2Institute of Biochemistry, Medical Faculty, University of Leipzig, 04103 Leipzig, Germany; 3Institute of Laboratory Medicine, Clinical Chemistry and Molecular Diagnostics, Leipzig University Hospital, 04103 Leipzig, Germany

**Keywords:** cancer, urine, cfDNA, mutation, liquid biopsy, T790M, digital PCR

## Abstract

**Highlights:**

-Model system using urine spiked with synthetic cfDNA reference standard to compare various urine-stabilizing buffers under different storage conditions in analogy to the clinical setting.-Direct preservation with UAS showed the best results, ensuring sufficient ucfDNA quality for downstream analysis.-Specific detection of the EGFR variant T790M using ddPCR.

**Abstract:**

Background: The “Liquid Biopsy” has become a powerful tool for cancer research during the last decade. Circulating cell-free DNA (cfDNA) that originates from tumors has emerged as one of the most promising analytes. In contrast to plasma-derived cfDNA, only a few studies have investigated urinary cfDNA. One reason might be rapid degradation and hence inadequate concentrations for downstream analysis. In this study, we examined the stability of cfDNA in urine using different methods of preservation under various storage conditions. Methodology: To mimic patient samples, a pool of healthy male and female urine donors was spiked with a synthetic cfDNA reference standard (fragment size 170 bp) containing the T790M mutation in the EGFR gene. Spiked samples were preserved with three different buffers and with no buffer over four different storage periods (0 h; 4 h; 12 h; 24 h) at room temperature vs. 4 °C. The preservatives used were Urinary Analyte Stabilizer (UAS, Novosanis, Wijnegem, Belgium), Urine Conditioning Buffer (UCB, Zymo, Freiburg, Germany) and a self-prepared buffer called “AlloU”. CfDNA was extracted using the QIAamp MinElute ccfDNA Mini Kit (Qiagen, Hilden, Germany). CfDNA concentration was measured using the Qubit™ 4 fluorometer (Thermo Fisher Scientific, Waltham, MA, USA). Droplet digital PCR (ddPCR) was used for detection and quantification of the T790M mutation. Results: Almost no spiked cfDNA was recoverable from samples with no preservation buffer and the T790M variant was not detectable in these samples. These findings indicate that cfDNA was degraded below the detection limit by urinary nucleases. Stabilizing buffers showed varying efficiency in preventing this degradation. The most effective stabilizing buffer under all storage conditions was the UAS, enabling adequate recovery of the T790M variant using ddPCR. Conclusion: From a technical point of view, stabilizing buffers and adequate storage conditions are a prerequisite for translation of urinary cfDNA diagnostics into clinical routine.

## 1. Introduction

Liquid Biopsy from body fluids like blood, urine, cerebrospinal fluid (CSF), saliva, or bone marrow has become a powerful tool in cancer research and diagnostics in recent years [1]. It can be applied alternatively or complementarily to traditional tumor tissue biopsy [2,3]. Liquid biopsies provide clinical information that is easy to obtain while bearing minimal inconveniences and risk to the patient. Particularly, urine collection—compared to a blood draw—is a truly non-invasive and painless method that could be used for long-term measurements to study the dynamics of a malignant disease and the efficacy of treatments [4,5]. In addition, large volumes can be acquired easily.

One of the most interesting analytes in Liquid Biopsy is circulating cell-free DNA (cfDNA), which has the potential to serve as a future biomarker for cancer [6]. Tumor-derived cfDNA, so-called ctDNA, contains valuable information concerning genetic and epigenetic alterations of the tumor. Sources of cfDNA in blood as in urine are apoptotic or necrotic cells, which might be associated with tumor development and resistance mechanisms [7]. There is the hypothesis that tumor cells actively and spontaneously release ctDNA into the bloodstream [8]. Particularly during metastasis, increased ctDNA levels in the circulating system have been described [5]. The total level of cfDNA includes both ctDNA and cfDNA from surrounding tissues and peripheral cells [9]. Dying non-malignant cells during tumor growth might cause increased cfDNA levels due to shed DNA. CtDNA only represents a small part (0.1–10%) of the total cfDNA [10]. Still, cfDNA levels of cancer patients were reported to be increased compared to healthy individuals [11]. Increased cfDNA concentrations are not only cancer-specific but can also appear in other diseases or during pregnancy, physical exercise and physiological stress; thus, measuring the concentration alone is not sufficient. Furthermore, gender-related differences in cfDNA yields have been observed [12]. A diagnostic impact was attributed to the mutational profiling of cfDNA, e.g., with next-generation sequencing (NGS) or droplet digital PCR (ddPCR) [7]. However, for disease monitoring over time, it might still be valuable to compare cfDNA levels. The most frequently occurring size of DNA fragments in plasma was 166 bp (the size which results from ”wrapped around a nucleosome”) or multiples of it, which are known to be characteristic for apoptotic cell death [13]. Further, DNA fragments larger than 10,000 bp were observed, which corresponded to necrotic cell death [7,14]. Other studies found DNA fragments between 150 and 250 or 400 bp in ucfDNA [15,16]. Interestingly, ctDNA derived from tumors was generally shorter (~145 bp) compared to non-tumor cfDNA [17]. Urinary cfDNA either originates directly from the genitourinary tract cells or from cfDNA that already circulated in the blood and passed the glomerular filtration, also called transrenal DNA [4,18]. The pore size of the glomeruli leads to a “dimensional selection”, as only small circulating fragments of about 100 bp are allowed to pass through [19]. Therefore, longer fragments are likely to originate from the genitourinary tract cells. Only about 0.5–2.0% of cfDNA in the blood circulation crosses the kidney barrier [4]. Therefore, large volumes of urine might be required to obtain a sufficient amount of cfDNA in a sample. The analysis of urinary ctDNA (uctDNA), however, has a variety of potentially valuable clinical applications, especially in the field of oncology: it might be useful as a screening tool, for cancer progression surveillance, for prognosis prediction and for therapy response monitoring. Relevant markers could be the concentration of total ucfDNA, tumor-specific mutations in uctDNA or ucfDNA integrity [18]. Since ucfDNA predominantly consists of physiological urothelial DNA, very sensitive assays are required to detect ctDNA against a high background of physiological ucfDNA [20]. Despite the advantages of urine sampling, the main obstacle remains the stabilization of ucfDNA for sufficient downstream extraction. In urine, the half-life time of cfDNA is shorter than in plasma due to high levels of nucleases like DNase I and DNase II, as well as contaminants present in urine [9,10]. In previous studies, we were able to show that targeted sequencing of ucfDNA from patients with breast cancer was feasible without using stabilizing buffers when samples were processed within 2–4 h after collection [11,21]. However, the cfDNA yield from 10 mL of urine appeared to be low. Therefore, stabilization of ucfDNA might be necessary in order to obtain adequate cfDNA levels for downstream analysis. In this study, we investigated the stability of cfDNA in urine. Additionally, we compared different storage conditions such as various stabilizing buffers, temperatures and several time periods as they occur in the clinical setting. The aims of our study were (1) to establish an adequate model system that mimics patient samples and real-world storage conditions as they occur during clinical routine, (2) to investigate the performance of various cfDNA-stabilizing reagents in urine samples under the mimicked conditions using fluorometry, and (3) to quantify and analyze the recovery of spiked cfDNA in stored urine samples using ddPCR.

## 2. Material and Methods

### 2.1. Synthetic cfDNA Reference Standard with T790M

To establish the experimental setup and to model clinical samples, we used the Multiplex I cfDNA Reference Standard Set in Synthetic Plasma (Horizon Discovery, Cambridge, UK), which contained DNA fragments with a size of 170 bp and covered eight onco-relevant mutations in the genes EGFR, KRAS, NRAS and PIK3CA with different allelic frequencies (AFs) of 0.1%, 1%, 5% and wild-type (wt). We focused on the T790M mutation of the epidermal growth factor receptor (EGFR) gene. EGFR, a tyrosine kinase, is overexpressed in stomach, brain and breast tumors. There are multiple mechanisms of EGFR dysregulation. Overexpression of the receptor might lead to the hyper-responsiveness of cancer cells to growth factors and was shown to trigger tumor proliferation and invasiveness [22,23]. Tyrosine kinase inhibitors can be applied for personalized therapies, but some cancer cells are able to develop resistances against these inhibitors [24]. The most common EGFR variant is T790M, a point mutation in exon 20 where threonine is substituted for methionine in amino acid position 790 of the EGFR gene domain [25,26].

### 2.2. Collection and Preparation of Urine Samples from Healthy Donors

For sample preparation, urine samples from four male and four female healthy volunteers were collected, each contributing a volume of 60–90 mL. Urine was pooled and experiments were set up using aliquots thereof in order to provide the same conditions for each sample and to assure comparability. Before combining the collected urine, triplicates with 3 mL each were aliquoted from each individual specimen and stored at −80 °C for later use. After mixing thoroughly by pipetting up and down, the urine pool was split into 3 mL aliquots. For the experimental part, each condition was set up in triplicates.

### 2.3. Choice of Urine Preservation Buffers for cfDNA Stabilization

We decided to test three different buffers. The “Urinary Analyte Stabilizer” system (UAS, Novosanis, Wijnegem, Belgium), in contrast to other buffers, involves direct transfer of the sample into the buffer by the patient, enabling immediate stabilization for reportedly up to 30 days at room temperature [27]. The “Urine Conditioning buffer” (UCB, Zymo Research, Freiburg, Germany) has shown promising results in previous studies [28]. The “AlloU” buffer was established by Zhou et al. and reportedly stabilizes ucfDNA over a period of 5 days. We prepared the AlloU buffer ourselves according to the main published components [29]. Briefly 2.5 g diazolidinyl urea (Thermo Fisher Scientific, Waltham, MA, USA), 2.5 g imidazolidinyl urea (Thermo Fisher Scientific, Waltham, MA, USA) and 2.5 g EDTA (Carl Roth, Karlsruhe, Germany) were combined. De-ionized water was added to a total volume of 50 mL and mixed on a magnetic stirrer for 10 min. Then, it was adjusted to pH 8 with sodium hydroxide (Carl Roth, Karlsruhe, Germany). The buffer was applied at a final concentration of 4%, which equals 120 µL in 3 mL of urine. AlloU consists of EDTA (ethylenediaminetetraacetic acid), diazolidinyl urea and imidazolidinyl urea, as well as Tris-HCl (Tris-(hydroxymethyl)-aminomethane hydrochloride, Merck, Darmstadt, Germany) as a pH buffer. EDTA acts as a nuclease inhibitor because it chelates divalent metal ions [30]. DNAzymes often present divalent cations like Zn^2+^, Mg^2+^ or Ca^2+^ as unique cofactors [31]. As strong Lewis acids, these metal ions can stabilize various secondary DNA structures and bind to both the negatively charged phosphate backbone and various nucleobases [32]. Therefore, metal ions are important players in catalysis mechanisms of nucleases and targets for nuclease inhibitors like EDTA. The role of diazolidinyl urea and imidazolidinyl urea in the urine preservation buffer is their antimicrobial effect [33,34]. The buffers UAS and UCB were purchased and then prepared according to the manufacturer’s instructions.

### 2.4. Experimental Sample Setup to Mimic a Patient Sample during Clinical Routine

The setup included four different storage periods: 0 h, 4 h, 12 h and 24 h. They represent varying clinical scenarios: immediate sample processing, a moderate 4 h delay and, because it was forgotten, processing later during the day or even the following day. Further, the entire setup was applied at two different temperatures: room temperature (RT) and fridge temperature (4 °C). The volume of the cfDNA reference standard solution was 60 µL for each spike-in, which equaled 24 ng of cfDNA in 3 mL of urine. For storage periods over 0 h, 4 h and 24 h, the reference cfDNA standard with 5% AF was used, as these samples would be applied to subsequent mutation detection using ddPCR (Table 1). After respective time periods, the tubes containing +/− buffers and +/− cfDNA reference standard were centrifuged at 3000× *g* for 10 min in order to pellet and remove cells. The supernatant was transferred into a clean DNAse-free tube (Thermo Fisher Scientific, Waltham, MA, USA), and then frozen at −80 °C until further processing.

Additionally, a positive control was run using 3 mL of distilled water instead of urine and a wild-type spiked-in cfDNA reference of 60 µL each. For the UCB, one additional setup without centrifugation of the sample was carried out. Urine of only one healthy individual was used and aliquoted into 3 mL triplicates. UCB and reference cfDNA, except for the negative controls, were added and the samples were stored at RT and 4 °C, respectively.

### 2.5. Isolation of cfDNA from Model Urine Samples

The experimental urine samples were thawed for one hour at RT. Prior to cfDNA isolation from urine, high g-force centrifugation (16,000× *g*) was necessary to separate the cell debris from the cell-free supernatant in order to reduce contamination. Several methods and kits are available for cfDNA extraction from body fluids such as plasma and urine. Most of them are based on either spin columns or magnetic beads. Based on our previous studies [11,21], we used the QIAamp MinElute ccfDNA Mini Kit (Qiagen, Hilden, Germany), according to the manufacturers’ instructions. Briefly, pre-concentration of cfDNA on magnetic beads was followed by purification using silica-based spin columns. CfDNA was then eluted into 25 µL of ultraclean water. This elution volume was suitable to obtain enough volume for the subsequent fluorometric cfDNA quantification, and to achieve an adequate concentration required for ddPCR analysis in duplicates. To maximize the DNA yield, the eluate was re-applied to the column as recommended in the manufacturers’ handbook. Resulting cfDNA concentrations were measured immediately after elution.

### 2.6. Fluorometric Quantification of ucfDNA

For quantitation of cfDNA yield, we used the Qubit™ 4 Fluorometer (Thermo Fisher Scientific, Waltham, MA, USA). The fluorometric assay was based on fluorescent dyes that emit only when bound to the DNA. In the dsDNA (double-stranded DNA) HS (High Sensitivity) Assay Kit (Invitrogen™; Thermo Fisher Scientific, Waltham, MA, USA), the fluorescent dyes bind selectively to double-stranded DNA, allowing sensitive detection of low DNA concentrations in the range of 0.1–120 ng per assay. For DNA quantification, a working solution was prepared using the dsDNA HS Assay Kit. The Qubit™ dsDNA HS Reagent was diluted 1:200 in the buffer. The kit contained two standard solutions and 10 µL of each standard was mixed with 190 µL of the working solution. After fluorometric reading, a calibration curve was obtained. To read the samples, 5 µL of the eluate was mixed with 195 µL of the working solution. Each sample was read three times.

### 2.7. ucfDNA Quantification and Detection of T790M Variant Using ddPCR

An alternative and more precise DNA quantification method is ddPCR, which enables mutant allele detection in a background of wild-type DNA [35]. DdPCR was performed using the QX200 Droplet Digital PCR System (Bio-Rad, Hercules, CA, USA) and the EGFR T790M assay (dHsaMDV2010019) (Bio-Rad, Hercules, CA, USA), according to the manufacturers’ protocol. Each reaction contained 7 µL of the sample and 15 µL of 2× ddPCR SuperMix for probes. The final reaction mix had a volume of 22 µL per sample and was transferred into a 96-well plate, followed by droplet generation on an automated droplet generator, and was then sealed with a pierceable foil (Bio-Rad, Hercules, CA, USA). The PCR was performed using C1000 Touch™ thermal cycler (Bio-Rad, Hercules, CA, USA). The following thermocycling conditions were chosen [26]: an initial incubation at 95 °C for 10 min, 47 cycles of 94 °C for 30 s, 57 °C for primer annealing for 1 min and finally inactivation at 98 °C for 10 min. The ramp rate was 2 °C/s. After cycling, the plate was incubated in the cycler at 11 °C overnight. Droplets were analyzed on a QX200 droplet reader (Bio-Rad, Hercules, CA, USA). Fluorescence was measured in two channels: FAM channel for mutant DNA, and HEX channel 2 for wild-type. Droplet separation allowed absolute quantification of the target and wild-type DNA and thus the sensitive detection of mutations present in the sample [36]. QuantaSoft™ Software Version 1.7 Regulatory Edition (Bio-Rad, Hercules, CA, USA) enabled the quantification of positive and negative droplets and, using Poisson statistics, determination of wild-type and mutant DNA concentrations [35]. Based on that, fractional abundance (FA) of the mutant molecule in the wild-type DNA was calculated using the software. Based on the literature, an FA of up to 0.001% may be detected using ddPCR [36]. Each sample was prepared and generated in duplicates. No-template controls (NTCs) were used in order to exclude possible contaminations. For positive-template controls, cfDNA reference standard solutions were used with the 5%, 1% and 0.1% AF.

### 2.8. Statistical Analysis

Concentrations of cfDNA were calculated as the mean value of each triplicate as well as the standard deviation. Student’s *t*-test was used to compare cfDNA differences between conditions. The level of significance was set to *p* ≤ 0.05. The results from ddPCR were analyzed using QuantaSoft™ software Version 1.7 Regulatory Edition (Bio-Rad, Hercules, CA, USA). The concentration (copies/μL) and the AF of each sample duplicate were calculated as mean value.

## 3. Results

### 3.1. Preliminary Tests

#### 3.1.1. Volume of Reference Standard

To establish the amount of cfDNA reference standard required to mimic a patient sample, 60 µL (24 ng) and 120 µL (48 ng) were tested. Preliminary tests showed that without any stabilizing buffer, the cfDNA yield (measured fluorometrically) was 17.5% (4.20 ng) using 60 µL and 23.9% (11.47 ng) using 120 µL of the reference standard. With the UAS buffer, about 50% of the cfDNA reference standard was recovered for both volumes. Therefore, 60 µL was chosen as the volume for spike-in experiments.

#### 3.1.2. Selection of Elution Volume

The QIAamp MinElute ccfDNA Mini Kit allowed the final elution volume to be chosen between 20 and 80 µL of ultraclean water. The elution volume should be low as ddPCR requires distinct DNA concentrations for proper results. Preliminary tests were run comparing 20 µL and 50 µL using the UAS buffer as the cfDNA stabilizer and 48 ng of the reference standard. cfDNA was extracted after 0 h and 2 h. For the measurements with the Qubit™, 5 µL was used for the 20 µL elution volume and 20 µL was used for the 50 µL elution volume. Using 50 µL as the elution volume revealed higher DNA yields compared to 20 µL as the elution volume (26.25 vs. 19.19 ng), but after 2 h, there was a loss of about 5 ng (21.54 vs. 18.55 ng). With the elution volume of 20 µL, more stability of the DNA yield could be observed over time; hence, 20 µL was preferred over 50 µL. Adding an additional 5 µL for fluorometric quantification, a total elution volume of 25 µL was used for all further extractions.

#### 3.1.3. Concentration of Buffer AlloU

The final concentration of buffer AlloU established by Zhou et al. was 2%, which is 60 µL in 3 mL of urine. For testing different concentrations, the double and tenfold (4%, 120 µL and 20%, 600 µL, respectively) were used as well. The recovery from 24 ng of the spiked-in reference standard was 7.76 ng with 2%, 10.59 ng with 4% and 9.49 ng with 10% buffer. Thus, a 4% AlloU urine concentration was applied.

### 3.2. Analysis of ucfDNA in Healthy Urine Samples

Eight different urine samples were analyzed individually before pooling. Sample donors included four healthy females (F1–F4) and four healthy males (M1–M4). The cfDNA was extracted from the pure samples of 3 mL (triplicates), without using any spike-in, and the “natural” cfDNA concentration was measured fluorometrically (Table 2). The individual cfDNA levels varied among the donors and ranged from not detectable (<0.1 ng) in M1 to a mean of 73.53 ng in F2. Less than 3 ng in 3 mL of urine was obtained in 5 out of 8 individuals. There was no significant difference between ucfDNA levels in males and females (*p* = 0.15). The mean value of all samples was 12.56 ng in 3 mL of urine (mean females: 19.75 ng; mean males: 5.37 ng; Figure 1).

### 3.3. Analysis of Model Urine Samples Spiked with cfDNA Reference Standard and Matching Negative Controls

For spike-in experiments, 24 ng of the cfDNA standard was used for each sample. For each condition, a negative control (NC) without artificial cfDNA was run using the same settings (buffer, time, temperature) as the matching spiked-in sample. The different buffers elicited various cfDNA dynamics throughout the chosen storage conditions (Figure 2).

#### 3.3.1. Storage without Preservation Buffer

CfDNA levels in the NC were significantly higher compared to the spiked samples throughout the entire time course (RT: 0 h: *p* = 0.022, 4 h: *p* = 0.015, 12 h *p* = 0.007, 24 h: *p* = 0.004; 4°C: 12 h: *p* = 0.049, 24 h: *p* = 0.017; Figure 2A; Appendix A). Calculating the quantity of the recovered cfDNA reference material that remained after DNA isolation by subtracting the NC from the spiked samples would result in negative values. Hence, cfDNA dynamics were investigated in both NC and spiked samples separately. Compared to the starting point (0 h), the cfDNA level in the NC increased after 4 h (*p* = 0.044) at RT, remained in this range over the duration of 12 h (*p* = 0.646) and decreased after 24 h (*p* = 0.071). The cfDNA level did not significantly differ between 0 h and 24 h in the NC (*p* = 0.698) stored at RT. Similarly, in the spiked sample, cfDNA dynamics revealed a slight but not significant increase after 4 h (*p* = 0.264) compared to the concentrations at 0 h, remained in this range at 12 h (*p* = 0.109) and showed a significant drop after 24 h compared to the level after 12 h at RT (*p* = 0.036). The cfDNA level did not significantly differ between 0 h and 24 h at RT in the spiked sample with no preservation buffer (*p* = 0.204). In contrast, when storing the NC samples at 4 °C, cfDNA levels remained stable during the first 12 h, but were significantly increased after 24 h (*p* = 0.002). In the spiked samples, we revealed elevated cfDNA levels after 4 h (*p* = 0.008) which remained stable until 12 h (*p* = 0.706) and were significantly increased after 24 h at 4 °C without preservation (*p* = 0.022; Figure 2A; Appendix A).

#### 3.3.2. Storage with UAS

Applying the UAS (Figure 2B), cfDNA analysis revealed significantly higher levels in the spiked samples compared to the NC at all conditions—as expected (*p* = 0.001). The relative yields of recovered cfDNA reference material ranged between 5 and 11 ng when subtracting the NC from the spiked samples (Appendix A), which was almost half of the spiked-in DNA (24 ng). Up to 12 h, cfDNA levels remained stable in the NC at RT (*p* = 0.136 at 4 h; *p* = 0.558 at 12 h) and were increased after 24 h (*p* = 0.008). It is worth noting that the cfDNA levels in the spiked samples increased between 0 h and 4 h at RT (*p* = 0.053), showed a drop after 12 h (*p* = 0.038), returned back to the initial level (*p* = 0.558) and were significantly elevated at 24 h (*p* = 0.001). When subtracting the NC from the spiked condition, the relative cfDNA yield remained stable, though with only a non-significant dip at 12 h (0 h: 7.45 ng cfDNA; 4 h: 8.20 ng cfDNA, *p* = 0.488; 12 h: 5.23 ng cfDNA, *p* = 0.065; Appendix A). After 24 h at RT, the total cfDNA level (9.12 ng) was not significantly different from the level at 4 h or 0 h (*p* = 0.248 or *p* = 0.169, respectively) in the spiked samples. In comparison, when storing the NC samples at 4 °C, cfDNA levels decreased within 4 h (*p* = 0.014), remained stable up to 12 h (*p* = 0.176) and were increased after 24 h (*p* = 0.029). The spiked samples remained stable at 4 h (*p* = 0.214) and 12 h (*p* = 0.394), but when subtracting the decreased NC, the remaining relative cfDNA yield appeared to be increased from the initial 7.45 ng to 10.43 ng after 4 h (*p* = 0.027), not after 12 h, though (*p* = 0.161; 8.32 ng cfDNA; Appendix A). After 24 h of storage at 4 °C, the cfDNA level was still not significantly different from the levels at 12 h (*p* = 0.487) or at 0 h (*p* = 0.474) even when the decreased NC levels were subtracted (*p* = 0.248), indicating that UAS preserved the reference material well.

#### 3.3.3. Storage with AlloU

Using the AlloU buffer (Figure 2C), cfDNA concentrations in the NC were decreased after 4 h at RT (*p* = 0.012), remained low up to 12 h (*p* = 0.909) and were increased after 24 h (*p* = 0.001). In the spiked samples stored at RT, cfDNA levels were significantly reduced after 4 h (*p* = 0.007) and remained low after 12 h (*p* = 0.451) and 24 h (*p* = 0.850). Although the initial relative cfDNA yield appeared to be promising after deducting the NC (0 h: mean 11.9 ng with highest value 13.42 ng of 24 ng), nearly half of the cfDNA reference was degraded after 4 h (mean: 6.65 ng; *p* = 0.009) and was further decreased after 12 and 24 h (5.5 and 3.8 ng; Appendix A). When storing AlluO-buffered NC samples at 4 °C, cfDNA remained stable up to 4 h (*p* = 0.245) and decreased after 12 h (*p* = 0.017). After 24 h, the levels were elevated (*p* = 0.015) but not significantly different from the cfDNA levels at 4 h (*p* = 0.7329). Similarly, cfDNA dynamics of the spiked samples remained stable up to 4 h (*p* = 0.533), dropped at 12 h (*p* = 0.015), but were elevated at 24 h (*p* < 0.001). Levels at 0 h, 4 h and 24 h were not significantly different (*p* = 0.533 and *p* = 0.997). The relative cfDNA yield after NC subtraction showed that the level remained stable with mean values around 11 ng and dropped down to 6.95 ng at 12 h (Appendix A).

#### 3.3.4. Storage with UCB

Employing UCB (Figure 2D) revealed that cfDNA concentrations in the NC were slightly increased after 4 h (*p* = 0.062) and remained in this range over the time course at RT (*p* = 0.855 after 24 h). In the spiked samples, the cfDNA levels were decreased by nearly 50% after 4 h at RT (*p* = 0.033). They were further reduced after 12 h (*p* = 0.018) and even lower than those of the NC, resulting in a negative relative cfDNA yield (Appendix A). Storage at 4 °C resulted in low cfDNA levels in the NC samples which did not differ after 24 h (*p* = 0.486). In the spiked samples, cfDNA concentrations dropped significantly after 4 h at 4 °C (*p* = 0.05), remained low up to 24 h (*p* = 0.576) and resulted in negative values when deducting the NC.

### 3.4. Relative cfDNA Dynamics in Spiked Urine Samples

The relative cfDNA dynamics after subtraction of the NC from each condition indicated that preservation with UAS and AlluO resulted in a higher recovery of the spiked reference material compared to UCB or no buffer (Appendix A). At RT, the highest yields were achieved with UAS (9.12 ng/3 mL; Figure 3A), and at 4 °C, AlluO resulted in the highest cfDNA levels after 24 h (11.21 ng/3 mL; Figure 3B). Storage with UCB revealed a significantly lower initial relative cfDNA yield already at 0 h compared to UAS (*p* = 0.004) and AlluO (*p* ≤ 0.001). The relative yield without any buffer was negative (Appendix A).

### 3.5. Observation of cfDNA Dynamics in NC Samples

Finally, we compared cfDNA dynamics in the NC samples (without synthetic reference cfDNA) under all four preserving conditions and at both temperatures (Figure 4). The NC of UAS and of no-buffer samples showed similar dynamics over time, with total DNA levels ranging from 6.41 ng/3 mL (no buffer, 0 h) to 15.90 ng/3 mL (UAS, 24 h RT; Appendix A). Using UAS, the natural cfDNA levels in healthy urine increased over time (*p* < 0.001 comparing 0 h vs. 24 h). After 24 h at RT, cfDNA levels were significantly elevated compared to samples stored at 4 °C (*p* < 0.001) and compared to samples stored with no buffer at RT (*p* < 0.001). With no buffer, the NC stored at RT revealed significantly increased cfDNA levels after 4 h (*p* = 0.044), which were decreased after 12 h and 24 h close to the initial levels (*p* = 0.698, 0 h vs. 24 h). During storage with no buffer at 4 °C, cfDNA levels in the NC samples rose continuously from an initial mean of 6.41 ng/3 mL up to 13.40 ng/3 mL after 24 h (*p* = 0.002), which was significantly higher compared to cfDNA levels observed in the UAS-buffered sample at that time (mean 10.20 ng/3 mL; *p* = 0.037).

When employing UCB or buffer AlloU, the cfDNA dynamics of the NC samples were similar to those of UAS and no-buffer NC, but the values were significantly lower, compared to the latter two conditions (24 h AlloU vs. UAS, *p* < 0.001, at RT and 4 °C, respectively). NC samples stored at 4 °C with AlluO or UCB showed cfDNA levels ranging between means of 1.08 and 2.18 ng/3 mL with almost no variation over time, except for a drop after 12 h with AlloU (*p* = 0.017; Appendix A) as described above. When storing the NC samples with AlloU at RT, the drop occurred already after 4 h (*p* = 0.012). The cfDNA level remained stable after 12 h and was significantly elevated up to a mean of 3.66 ng/3 mL after 24 h (*p* < 0.001; Appendix A). The dynamics of cfDNA in NC buffered with UCB at RT were as monotonic as at 4 °C.

In addition to these experiments, a positive control was run with the same amount of reference cfDNA spiked into distilled water instead of urine, also as a triplicate. The mean cfDNA concentration recovered after 0 h with no buffer was 13.17 ng/3 mL, which was about 55% of the spiked-in cfDNA amount.

### 3.6. Investigating the Concentration of Urinary cfDNA Using ddPCR vs. Qubit

As cfDNA stabilization using buffer AlloU and UAS resulted in the highest concentrations, we selected these two buffers for this follow-up experiment. We prepared NC and spiked samples and analyzed the WT and T790M variant at 0 h, 4 h and 24 h for both storage at RT and 4 °C with ddPCR. Reference standard solutions (AF: 5%, 1% and 0.1%) served as positive controls. DNA concentrations (copies/µL) were measured in all samples with more than 10,000 generated droplets, and in spiked-in samples, we determined the fractional abundance (FA) of the mutation in the background of WT DNA. Only samples with more than three positive droplets in channel 1 were considered positive for the T790M mutation. All samples except for one resulted in sufficient droplet generation and separation. The concentration from copies/µL to ng/assay was calculated as follows: one copy equals one haploid genome equivalent. Hence, the number of haploid genome equivalents was obtained by multiplying the concentration in copies/µL by 22 µL, which was the total volume of the applied ddPCR mix [37]. One haploid genome equals about 3.3 pg or 0.0033 ng [38]. Thus, the quantity of cfDNA in ng/assay is the number of haploid genome equivalents multiplied by 0.0033 ng.
$$\frac{ng}{assay}=\frac{copies}{\mu L}\times 22\;\mu L\times 0.0033\;ng$$

When comparing the amount of cfDNA measured fluorometrically to the amount of cfDNA estimated using ddPCR, the values were very similar (Appendix A). However, for the pure reference standard, the measured DNA levels with ddPCR were <2.8 ng, which was the amount we had expected in a volume of 7 μL. In preliminary experiments using the Qubit™ fluorometer, we were able to confirm the concentration of 400 ng/mL as provided by the manufacturer, though.

### 3.7. Detecting the T790M Variant in Urinary cfDNA Using ddPCR

The T790M mutation was measured in the samples preserved with UAS after 0 h, 4 h and 24 h of storage at both temperatures. At RT, the FA was slightly lower compared to 4 °C, but overall, the values were stable over time (FA (RT vs. 4 °C)—0 h: 2.05%; 4 h: 2.0% vs. 3.2%; 24 h: 1.9% vs. 1.9% Figure 5). Notably, the FA was around 2%, which was less than half of the spiked AF (5%). In the AlloU-buffered samples, T790M was present after 0 h, 4 h and 24 h of storage at 4 °C. However, after 24 h at RT, the mutation was no longer detectable in samples with this buffer (FA (RT vs. 4 °C)—0 h: 2.95%; 4 h: 2.25% vs. 2.5%; 24 h: n.d. vs. 2.0%). At 4 °C, the values ranged between 2% and 3% FA. In samples preserved with UCB or no buffer, T790M was not detectable. Therefore, the fractional abundance (FA) could not be analyzed.

In the reference standard solutions, which were used as positive controls, the T790M variant with 0.1% AF could not be detected. In the solutions containing T790M with 1% AF with respect to 5% AF, the variant was detectable. It is worth noting that the determined FA was even higher than 1% and 5% in those solutions (Appendix A). However, in none of the urine samples was the expected FA of 5% measured, which would equal the AF of the reference standard used. Here, the negative control plays an important role.

The FA of a mutation can be altered by high levels of cfDNA in the NC of the sample, as it is calculated with the equation $FA=\frac{a}{a+b}$, where a = mutant and b = wild-type [37].

If cfDNA levels in the NC were high, like for the UAS-buffered samples, the FA would be lower than 5%. The expected FAs for UAS- and AlloU-preserved samples were calculated with the following equation: $FA=5\%\times (1-\frac{NC}{Spike-in}$).

The EGFR variant T790M in samples preserved with UAS had nearly the same FA measured using ddPCR compared to the calculated FA, which was slightly higher. The measured FA for the samples with buffer AlloU did not meet the calculated values, as they were much lower (Table 3).

## 4. Discussion

Preliminary tests already indicated that the synthetic cfDNA reference standard degraded very quickly in urine without the usage of stabilizing buffers. This might be due to the high activity of nucleases including DNase I, DNase II and phosphodiesterase I, which are suitable in an environment with a lot of divalent cations and a pH value of 5–7 in urine [16]. Remarkably, in our study, the cfDNA concentrations in unbuffered urine samples spiked with synthetic reference material were not only low as it was expected without using any preservative, but they were even lower compared to the negative controls containing no spike-in but only (low) “natural” cfDNA from healthy donors. This was contrary to all expectations, as even without any stabilization, the cfDNA concentration should have been slightly higher than that of the NC, or at least as high.

A potential explanation of this finding is the presence of residual cells or cell debris in the urine after the first centrifugation (3000× *g*), which were then lysed during the freeze–thaw cycle and released their DNA into the samples, leading to contamination with degraded DNA fragments. In further experiments, we found that immediate cfDNA extraction after sample collection led to lower cfDNA concentrations. Previous studies have highlighted the influence of centrifugation speed and time on the cfDNA yield of unbuffered urine samples; therefore, a double centrifugation protocol is recommendable before freezing the samples [39].

In our study, the most effective stabilization of the spiked cfDNA reference standard was achieved using the UAS buffer. However, the total spiked amount of 24 ng (in 3 mL urine) was not recovered in any of the experiments. We attribute this result to the limited extraction efficacy of the QIAamp MinElute ccfDNA Kit, which was approximately 50% for the cfDNA reference standard according to the manufacturers’ product information [40]. Our preliminary experiments confirmed that up to 50% of the spiked cfDNA amount could be recovered using the UAS buffer. Further, the positive control using distilled water instead of urine, to rule out nuclease activity, confirmed that only 55% of the reference standard could be recovered (after 0 h). Taken together, the UAS buffer enabled stable cfDNA recovery for up to 24 h at both temperatures. Although the exact composition of the buffer remains unknown, it probably contains a nuclease inhibitor and antimicrobial reagents to protect the integrity of the cfDNA.

The cfDNA dynamics of UAS-buffered unspiked (NC) urine samples differed from unspiked unbuffered urine samples: In UAS-buffered samples, the increase in cfDNA concentration over time was higher at RT compared to 4 °C. Most probably, cells were lysed faster at RT and the released nucleic acids were then protected from nucleases by the UAS. Accordingly, cfDNA concentrations in the NC samples with no buffer decreased at RT, as the released DNA was degraded by nucleases. Vice versa, cfDNA levels in the NC with the UAS buffer increased at RT, as the released DNA was protected from degradation. At lower temperatures, biochemical processes are usually slower due to Brownian motion [41]; hence, cell rupture in urine might occur slower and cfDNA levels might subsequently increase less at 4 °C compared to RT. Interestingly, the increase in unspiked cfDNA was also visible in spiked samples, indicating a contamination of the spiked samples with gDNA fragments over time.

Using the AlloU buffer, 47% of the spiked cfDNA was recovered after 24 h of storage at 4 °C. The drop at 12 h probably resulted from technical errors. In contrast, the concentrations achieved at RT were lower than expected. After 24 h, only around 16% of the spiked amount remained. It appears that the buffer might only sufficiently protect the spiked reference standard at 4 °C. Hence, we could not confirm the performance of this buffer as described by Zhou et al. claiming cfDNA stability over five days at room temperature [29]. We assume that the preparation of the buffer was either incorrect or that some components were missing in our preparation, as only the main components were published in the literature. Another possible reason might be that EDTA, which was the main protecting component, inhibited the nucleases more effectively at 4 °C. Still, EDTA seemed to work well as a nuclease inhibitor with diazolidinyl urea and imidazolidinyl urea as antimicrobial reagents, which is in line with other studies using EDTA and antibiotics such as penicillin–streptomycin to preserve cfDNA in urine or blood [28,35].

Interestingly, the cfDNA levels in the NC samples (without spiked cfDNA) using AlloU were much lower in comparison to no buffer or UAS. This indicated that almost no gDNA potentially present in the urine was recovered by cfDNA extraction. An explanation for this observation could be that certain components in the AlloU buffer might bind fragments of higher length and transfer them to the pellet during centrifugation. This theory remains to be verified in future experiments including an analysis of cfDNA fragment length over time. The shorter, spiked-in reference fragments of 170 bp remained in the supernatant as the expected amount could be recovered, at least at 4 °C. The cfDNA levels in the NC with AlloU remained low over time. Only after 24 h at RT was an increase recorded—potentially the result of cell lysis. By preventing urinary cell lysis, the AlloU might minimize the amount of gDNA and possible interference during subsequent cfDNA analysis. This might represent an advantage compared to UAS, which did not seem to prevent cell lysis. However, the buffer prepared for this experimental setup requires some optimizations for a better performance preserving cfDNA at RT.

The cfDNA yield using UCB was very low for both temperatures and all storage periods. According to the manufacturers’ protocol, the cellular and cell-free fraction cannot be separated after adding UCB to the sample [42]. The spiked cfDNA was probably transferred to the cell pellet during centrifugation; thus, almost no cfDNA was detected in the supernatant. Further, the NC revealed very low cfDNA levels, similar to the results obtained with AlloU. Therefore, a urine sample should be centrifuged and the cell pellet removed before UCB is added to the supernatant. This procedure, however, is not feasible in a clinical setting, because ucfDNA will likely degrade before centrifugation if no buffer is present, as we could show in our experiments. Nursing staff would need to centrifuge the urine immediately after sample collection, which is usually not possible during clinical routine. Even if it was, the cfDNA dynamics with no buffer showed that during 10 min of centrifugation, nucleases might already degrade the majority of cfDNA in the sample (Figure 2A). Alternatively, we added UCB straight to the urine sample and investigated whether cfDNA could be extracted without prior centrifugation. However, our test revealed that only one third of the spiked cfDNA was recovered after 0 h, and even less after 24 h of storage. The background of gDNA derived from lysed cells in the urine samples was very high and would most likely hinder any subsequent analyses such as rare mutation detection. This study showed that urine samples require preservation directly after collection in order to achieve sufficient cfDNA concentrations. Therefore, UCB is not a suitable buffer for rare event detection in ucfDNA. For investigation of bladder cancer, both the urine sediment and the supernatant might contain valuable information, but for other cancers, the gDNA in the sediment might contaminate ucfDNA and thus should be separated from the supernatant [43,44].

We used ddPCR to quantify recovered cfDNA levels and to confirm the synthetic origin by detecting the T790M variant. The ddPCR is considered a method with high sensitivity and specificity for Liquid Biopsy analysis [45] and was already used in many studies for the detection of specific mutations like the T790M [24,26]. It was reported that even AFs as low as 0.5% were detectable. In our study, we were able to detect the cfDNA reference standard with 1% AF very precisely. The standard with 0.1% AF was not detected, though. However, our results are in line with a study by Silveira et al., as the detection limit appeared to be somewhere between 0.1% and 1% [26]. We successfully showed that ddPCR was very specific for the T790M mutation. Furthermore, the majority of samples revealed similar cfDNA concentrations as measured fluorometrically in comparison to quantification using ddPCR. However, the cfDNA levels of the pure reference standard solutions measured using ddPCR were constantly lower than the actual concentration. Using the Qubit™ fluorometer, we could determine the concentration of 400 ng/mL (provided by the manufacturer) exactly. Remarkably, the differences between the cfDNA concentrations measured with both methods varied according to the stabilizing buffers used. The cfDNA quantity assessed using ddPCR was higher in the samples with no buffer or UAS compared to fluorometric quantification. Which one of both methods revealed the true value could not be determined conclusively. The elevated “natural” cfDNA levels observed in the NC samples (no spike-in) with no buffer and with UAS might be one reason for the increased calculated values.

The spiked T790M mutation on the EGFR gene was only detectable in samples buffered with UAS and AlloU, not in the samples with no buffer or UCB. As the cfDNA concentrations barely differed between NC and spiked samples in the latter two conditions, we conclude that most of the reference standard degraded or was lost during cfDNA extraction (UCB). Using the UAS buffer, the FA of the T790M variant as measured using ddPCR was almost identical to the expected (calculated) FA for all storage periods and temperatures. This suggests that the UAS buffer sufficiently preserved spiked reference cfDNA fragments over time and that the FA estimation using ddPCR was very accurate. Concerning the AlloU buffer, the measured FA was lower than expected, indicating an inferior cfDNA preservation. Although fluorometric quantification using the Qubit™ showed that cfDNA stabilization with AlloU was acceptable at 4 °C, the UAS seemed to be the better option when aiming at rare event detection such as a specific mutation.

For the first time, we systematically compared different urine-stabilizing buffers over time and at different temperatures. We were able to identify future challenges in the pursuit of standardization of ucfDNA collection and processing for downstream diagnostic procedures. Towards this goal, we mimicked patient samples and storage conditions as they occur during clinical routine. We were able to observe different ucfDNA levels under various conditions and found that direct preservation with UAS showed the best results, ensuring sufficient ucfDNA quality for downstream analysis such as the detection of a specific variant even after overnight storage at room temperature.

Yet, the question remains regarding how far this model is comparable to a patient sample. The synthetic reference material appeared to mimic the cfDNA of a cancer patient adequately. However, during the experimental set up, synthetic cfDNA fragments were not exposed to nucleases in urine for the buffered conditions, as the buffers were added to the sample prior to spiking. In a patient, ucfDNA is exposed to nucleases for several hours in the bladder, where it might degrade. Degradation might even start in the blood circulation where nucleases are present, too, and the concentration of individual variants in tumor patients might also vary strongly inter-individually, depending on various factors such as vascularization or metabolism of the respective malignancy. These points emphasize the importance of a urine preservative immediately after sample collection or the use of pre-manufactured urine tubes in order to avoid the in vitro degradation of nucleic acids.

Further, we confirmed that elevated levels of cfDNA are not necessarily indicative of malignancies but can occur in other disorders as well. In our study, one of the female volunteers had increased ucfDNA levels, but was later revealed to have a urinary tract infection. The increased ucfDNA was therefore likely derived from leukocytes or urothelial cells, which were lysed in the urinary tract. Some studies reported gender-specific differences in DNA yields [12], while others reported no significant difference [16]. Our study did not reveal significant gender-related differences. Due to the small sample size, this finding remains to be verified in future studies. Another aspect worth mentioning is the cfDNA recovery. This study pointed out that a maximum of 50% of a synthetic reference cfDNA could be recovered when using an adequate stabilizing reagent (UAS). However, it remains unclear how much of the actual cfDNA we might extract from a patients’ urine sample.

Here, pre-analytical variables such as the choice of extraction method should be considered carefully. In terms of clinical diagnostics, sophisticated analysis of cfDNA fragments, so-called fragmentomics, will play a key role in future studies.

## 5. Conclusions

Despite the above-mentioned limitations, our approach discovered that the stability of cfDNA in urine was very low and that the use of adequate stabilizing reagents appears essential for subsequent cfDNA analysis. The best performance was achieved by the UAS buffer, resulting in a stable cfDNA yield after storage for 24 h at room temperature and 4 °C. The spiked T790M mutation could be detected at all storage periods and both temperatures using ddPCR. Further, buffer AlloU showed good cfDNA-stabilizing properties over time, but only at 4 °C.

## Figures and Tables

**Figure 1 diagnostics-13-03670-f001:**
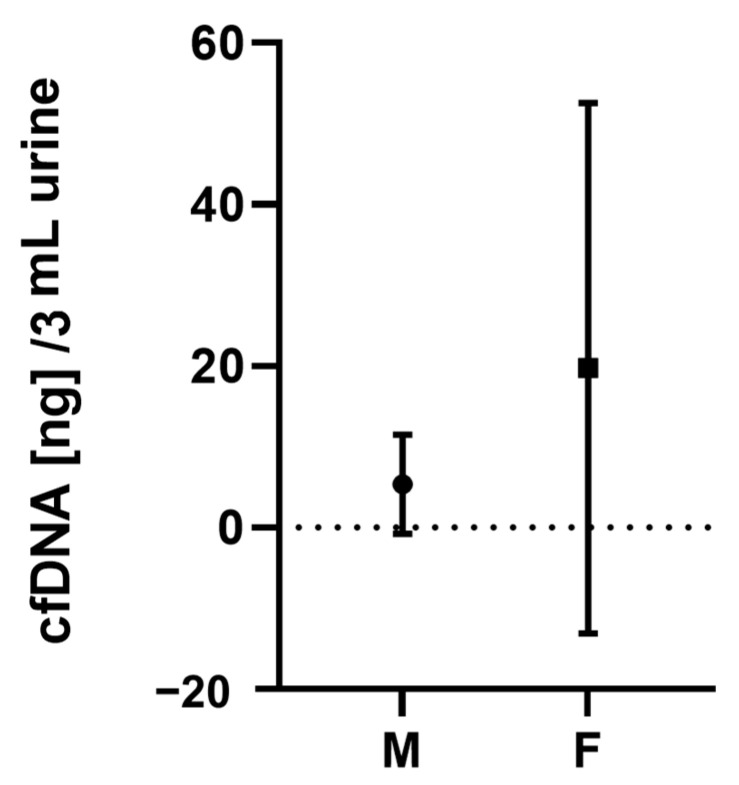
cfDNA levels in ng for female (F) vs. male (M) urine donors were not significantly different (*p* = 0.15; mean females: 19.75 ± 31.44 ng; mean males: 5.37 ± 5.85 ng).

**Figure 2 diagnostics-13-03670-f002:**
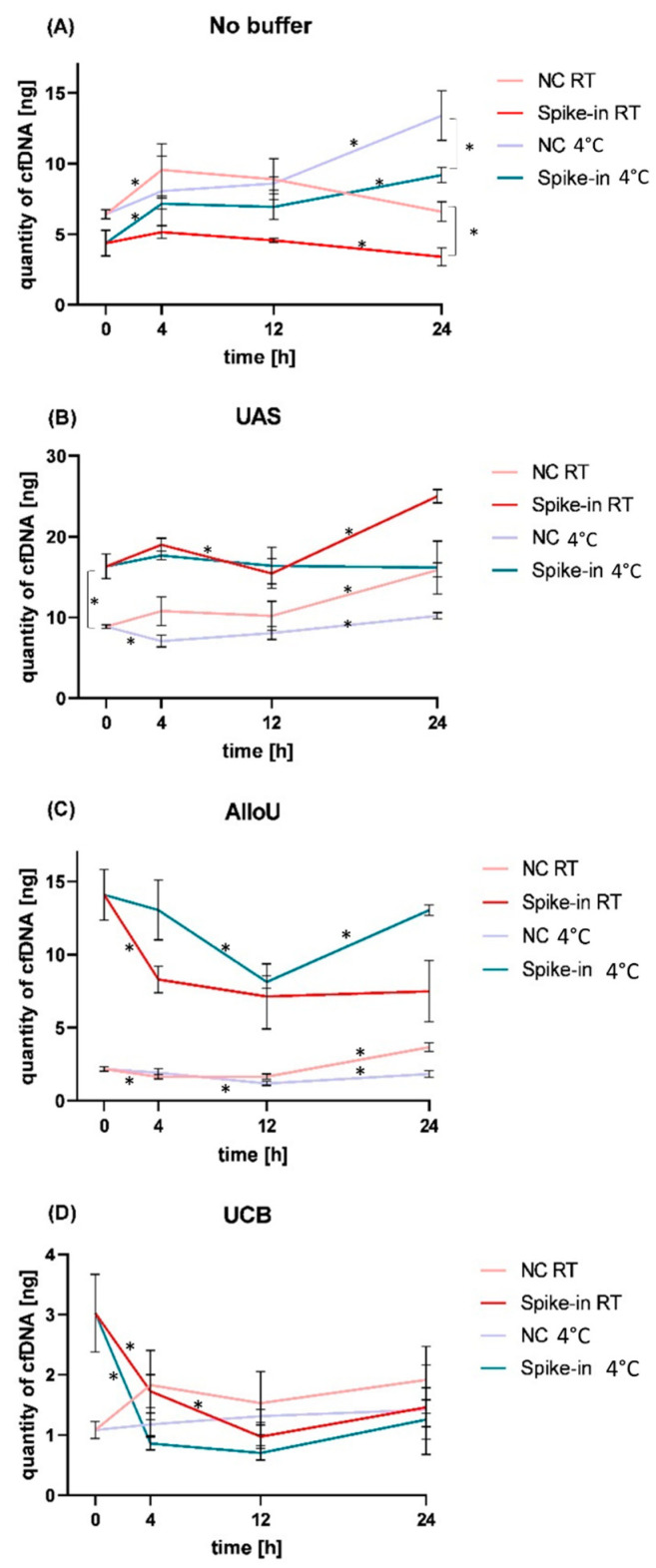
cfDNA dynamics in the NC and the samples spiked with cfDNA reference standard are depicted as mean values of total cfDNA content with their standard deviation for (**A**) no buffer, (**B**) buffer UAS, (**C**) buffer AlloU and (**D**) buffer UCB. The cfDNA is shown in ng after storage periods of 0 h, 4 h, 12 h and 24 h at room temperature (RT, red) and fridge temperature (4 °C, blue). The lighter colors show cfDNA levels of the NC. * indicates level of significance *p* ≤ 0.05 between total cfDNA content at consecutive time points.

**Figure 3 diagnostics-13-03670-f003:**
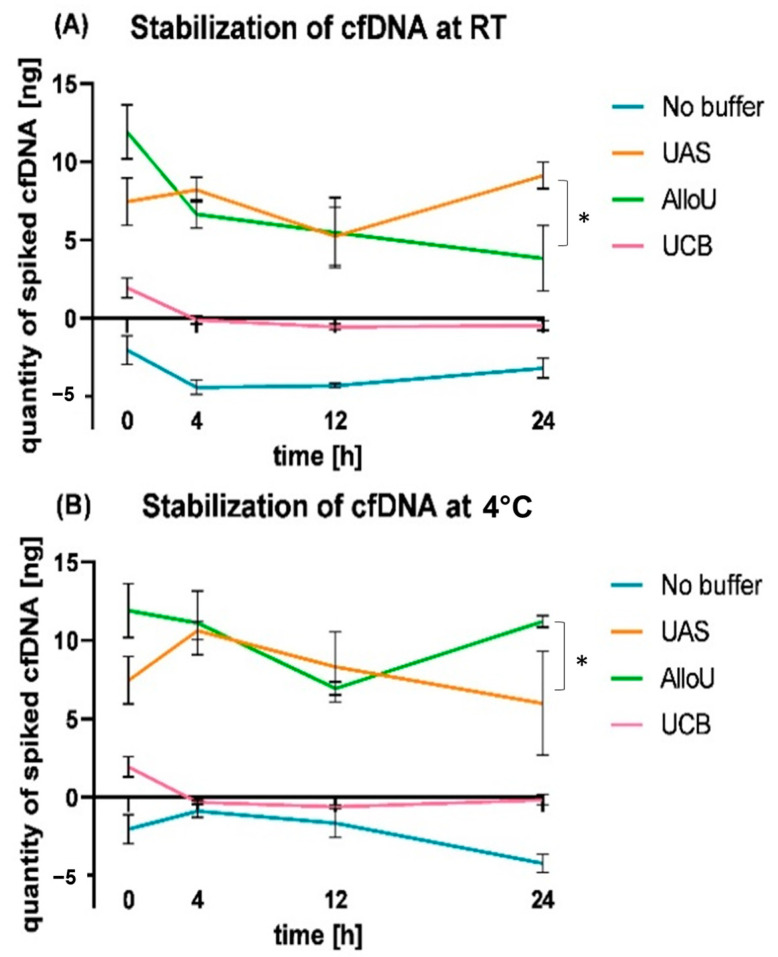
Comparison of relative cfDNA levels (ng) in four different preserving conditions after 0 h, 4 h, 12 h and 24 h at (**A**) room temperature (RT) and (**B**) 4 °C. The measured cfDNA levels in ng of the negative controls were subtracted from the cfDNA levels of the spiked samples. No buffer—blue, UAS—orange, AlloU—green, UCB—pink. After 24 h at RT, UAS resulted in the highest relative cfDNA yield of 9.12 ng from 3 mL urine, which was significantly higher compared to 3.82 ng recovered with AllUo (*p* = 0.015), but not significantly different from the initial level at 0 h (*p* = 0.169). Further, using AlloU revealed significantly decreased cfDNA levels after 24 h at RT (*p* = 0.007). After 24 h at 4 °C, however, UAS showed a relative cfDNA yield of only 4.28 ng, which was a non-significant decrease compared to the initial level (*p* = 0.248), but inferior to AlloU with 11.21 ng in 3 mL urine after 24 h (*p* = 0.034). * indicates level of significance *p* ≤ 0.05 between cfDNA levels.

**Figure 4 diagnostics-13-03670-f004:**
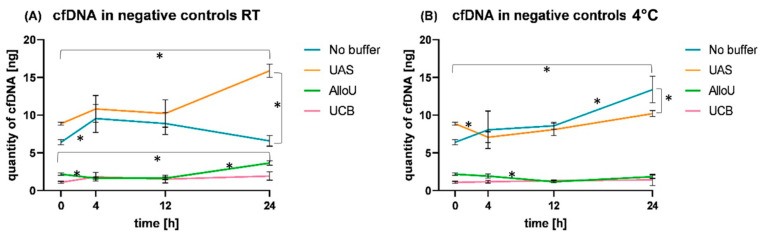
Comparison of cfDNA levels in the NC samples with the four conditions after 0 h, 4 h, 12 h and 24 h at (**A**) room temperature (RT) and (**B**) 4 °C. No buffer—blue, UAS—orange, AlloU—green, UCB—pink. * indicates level of significance *p* ≤ 0.05 between cfDNA levels.

**Figure 5 diagnostics-13-03670-f005:**
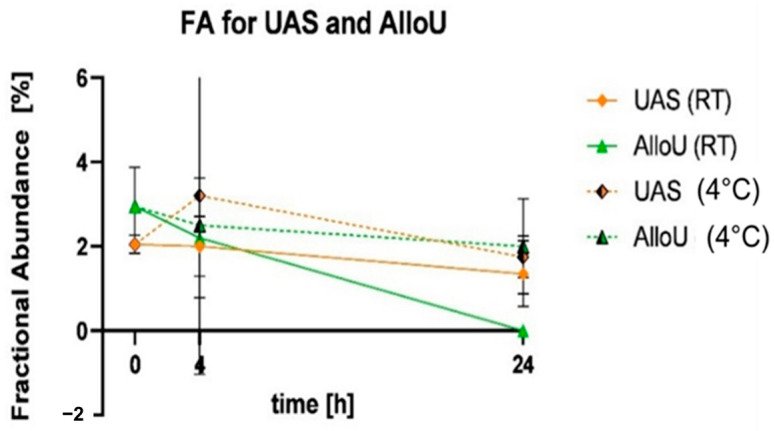
Fractional abundance (FA) of T790M mutation in samples buffered with UAS and AlloU. The samples were measured using ddPCR after storage periods of 0 h, 4 h and 24 h at room temperature (RT, solid lines) and 4 °C (dotted lines). All samples were spiked with cfDNA reference standard carrying the T790M variant at 5% AF. UAS—orange, AlloU—green.

**Table 1 diagnostics-13-03670-t001:** Pipetting scheme of sample setup. The negative control (NC) contained the same buffer (or no buffer) as the corresponding sample containing spiked-in cfDNA for a better comparison. For stabilization, 1 mL UAS buffer and 210 μL of UCB were used, according to the manufacturers’ instructions, respectively. Concerning buffer AlloU, a volume of 120 μL was used for 3 mL urine. To mimic patient samples, each 3 mL urine aliquot was spiked with 60 μL of the synthetic cfDNA reference standard solution, which equals 24 ng cfDNA.

Stabilizing Reagent	Sample Type	Urine Volume	Buffer Volume	Reference cfDNA Spike-In Volume
No buffer	NC	3 mL	-	-
No buffer	spike	3 mL	-	60 µL
U A S	NC	3 mL	1 mL	-
U A S	spike	3 mL	1 mL	60 µL
AlloU	NC	3 mL	120 µL	-
AlloU	spike	3 mL	120 µL	60 µL
UCB	NC	3 mL	210 µL	-
UCB	spike	3 mL	210 µL	60 µL

**Table 2 diagnostics-13-03670-t002:** Amount of cfDNA in ng/3 mL of urine obtained from female and male donors. Eight different healthy individuals donated 60–90 mL of urine, including four women (F1–F4) and four men (M1–M2). From each sample, triplicates with 3 mL each were taken and analyzed for their pure cfDNA level. The columns show the mean cfDNA concentration and standard deviation.

Sample	Total cfDNA/3 mL (ng)	Mean	SD	Level of Significance
F1	1.85			
	1.85	2.06	0.30	
	2.49			
F2	61.17			
	74.13	73.53	9.86	
	85.30			
F3	1.29			
	1.01	1.24	0.17	
	1.42			
F4	1.62			
	1.95	2.19	0.60	
	3.01			
**Female cfDNA (ng/3 mL)**		**19.76**	**31.44**	
M1	0			
	0	0	0	
	0			
M2	8.82			
	16.93	14.13	3.76	
	16.63			
M3	1.11			
	1.38	1.32	0.16	
	1.48			
M4	5.59			
	5.83	6.03	0.46	
	6.66			
**Male cfDNA (ng/3 mL)**		**5.37**	**5.85**	** *p* ** **= 0.15**

**Table 3 diagnostics-13-03670-t003:** Expected FA vs. measured FA of EGFR T790M for samples preserved with UAS and AlloU. The table compares the expected FA of the mutation, calculated with the ratio of cfDNA quantities of NC and spiked sample, with the FA estimated using ddPCR. The samples were measured for the time periods 0 h, 4 h and 24 h and both temperatures (RT, 4 °C).

Storage Conditions		Samples Buffered with UAS		Samples Buffered with AlloU	
Time (h)	Temperature	Expected FA of Variant (%)	Measured FA of Variant (%)	Expected FA of Variant (%)	Measured FA of Variant (%)
0		2.41	2.05	4.26	2.95
4	RT	2.41	2.00	3.95	2.25
4	4 °C	3.31	3.20	4.22	2.50
24	RT	1.96	1.90	2.65	0
24	4 °C	1.25	1.80	4.39	2.00

## Data Availability

Data is contained within the article and Appendix A.

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
