# Peer review of "The Challenge to Stabilize, Extract and Analyze Urinary Cell-Free DNA (ucfDNA) during Clinical Routine"

_diagnostics, 2023, doi:10.3390/diagnostics13243670_

Round 1

Reviewer 1 Report

Comments and Suggestions for Authors

Dear authors

The model system is nicely set up and all the analyses done very accurately. However, there remain many open questions in the end which are all outlined in the discussion.

Most intriguingly is the fact that more cfDNA is measured in NC samples compared to spike-in. Additionally the increase of cfDNA during time is not entirely clear to me. The authors comment it with remaining cells and debries after the centrifugation steps which release cfDNA during time. Thus, a more closer look and deeper investigation in this ‘problem’ might be of essential interest.

Interesting would as well be to see some TapeStation histograms/results. Especially from the initial donor samples, then after spike-in and in a time-course. This would give a quality parameter about the composition of debries vs. cfDNA (166bp) vs. spike-in (170bp) vs. ctDNA (145bp, if any there). As well it could be used for quantification of material (ng/ul).

Minor points:

line 84: The most frequently occurring size of DNA fragments in plasma was 166 bp (the size of a nucleosome) à rephrase since it is rather the size which results  from ‘wraped around a nucleosme’

line 91: …pore size of the glomeruli leads to a “dimensional selection”, as only 91 small circulating fragments of about 100 bp are allowed to pass through à is then the model system with spike-in EGFR p.T790M the right model system with 170bp since this mutation is mainly found in lung cancer as resistance mutation. Shouldn’t  we rather look then for <=100bp ctDNA fragments in future in urine? If yes then the Qiagen isolation method is the wrong one because the cut-off of the column is >100bp. You might outline this further in the discussion.

Author Response

Reviewer 1:

We thank the reviewer for thorough constructive comments, which we found very helpful.

Most intriguingly is the fact that more cfDNA is measured in NC samples compared to spike-in. Aditionally the increase of cfDNA during time is not entirely clear to me. The authors comment it with remaining cells and debries after the centrifugation steps, which release cfDNA during time. Thus, a more closer look and deeper investigation in this ‘problem’ might be of essential interest.

We thank the reviewer for this very important comment. The ocurence of increased cfDNA puzzled us just as much and we are considering further investigations.

Interesting would as well be to see some TapeStation histograms/results. Especially from the initial donor samples, then after spike-in and in a time-course. This would give a quality parameter about the composition of debries vs. cfDNA (166bp) vs. spike-in (170bp) vs. ctDNA (145bp, if any there). As well it could be used for quantification of material (ng/ul).

This is a very importat point. We apppreciate the reviewers advice concerning cfDNA quantification via fragment length in addition to our fluorometric method. In future experiments, we will apply electrophoresis to qualify and quantify nucleic acids in our model samples as well as in patient samples.  We added a sentence in the discussion.

line 84: The most frequently occurring size of DNA fragments in plasma was 166 bp (the size of a nucleosome) à rephrase since it is rather the size which results from ‘wraped around a nucleosme’

Many thanks for this detail. We rephrased the sentence accordingly.

line 91: …pore size of the glomeruli leads to a “dimensional selection”, as only small circulating fragments of about 100 bp are allowed to pass through à is then the model system with spike-in EGFR p.T790M the right model system with 170bp since this mutation is mainly found in lung cancer as resistance mutation. Shouldn’t we rather look then for <=100bp ctDNA fragments in future in urine? If yes then the Qiagen isolation method is the wrong one because the cut-off of the column is >100bp. You might outline this further in the discussion.

Point well taken. The cfDNA extraction is a highly relevant issue. The Multiplex I cfDNA Reference Standard Set in Synthetic Plasma (Horizon Discovery, Cambridge, UK) used in our study contains DNA fragments with the size of 170 bp and covers eight onco-relevant mutations in the genes EGFR, KRAS, NRAS and PIK3CA in the different allelic frequencies (AF) 0.1%, 1%, 5% and wildtype (wt). The focus in this study was on the T790M mutation on the epidermal growth factor receptor (EGFR) gene. EGFR, a tyrosine kinase, is overexpressed in stomach, brain, and breast tumors. This mutation is detectable with the ddPCR using a specifically designed primer (EGFR T790M assay kit; Bio-Rad, California USA), established in our laboratory. Hence, this method enabled the exact identification and quantification of the spiked cfDNA remaining under the chosen storage conditions. There are several methods and kits available on the market to extract cfDNA from body fluids such as plasma and urine. Most of them are based on either spin columns or magnetic beads. The QIAamp MinElute ccfDNA Mini Kit (Qiagen, Hilden, Germany) combines both methods by first preconcentrating the DNA on magnetic beads and then purifying it with silica-based spin columns. The use of magnetic particles for DNA separation bears many advantages: There are hardly any restrictions for the sample volume, magnetic particles can be removed easily and it is a very efficient method (Berensmeier 2006). Magnetic beads are coated with immobilized affinity ligands or a biopolymer with affinity to the target nucleic acid on their surface (Berensmeier 2006). After DNA binding, they can be separated in a magnetic field. Subsequently, cfDNA isolation can take place using a spin column. Qiagen’s spin columns are equipped with silica gel membrane technology. DNA binds to silica (amorphous silicon dioxide) in the presence of a chaotropic salt (buffer ACB). The principle for this mechanism is based on hydrophobic interactions and bonds between the phosphate group of the DNA and the silanole group of the silica membrane (Shi et al. 2015).  According to the manufacturers’ handbook the colums can extract fragments as short as 20 bases. Hence, for the model system used in our study, the MinElute column appeared to be an adequate choice.  

However, a recent study by Ruppert et al, used the same column and indicated decreased recovery of 64 bp fragments compared to fragemnts with a size of 127 bp  (Ruppert, T, Roth, A, Kollmeier, J, Mairinger, T, Frost, N. Cell-free DNA extraction from urine of lung cancer patients and healthy individuals: Evaluation of a simple method using sample volume up-scaling. J Clin Lab Anal. 2023; 00:e24984. doi:10.1002/jcla.24984). Therefore, additional analysis of fragment sizes will play an important role in future studies. We acepted this constructive criticism and added a passage at the end of the discussion section.

Reviewer 2 Report

Comments and Suggestions for Authors

Circulating cell-free DNA (cfDNA) that originates from tumors has emerged as one of the most promising analytes. In contrast to plasma-derived cfDNA, only a few studies have  investigated urinary cfDNA.

In this study, they  examined the stability of cfDNA in urine using different ways of preservation under various storage conditions. To mimic patient 14 samples, a pool of healthy male and female urine donors was spiked with a synthetic cfDNA reference standard  containing the T790M mutation in the EGFR gene. Stabilizing buffers showed varying efficiency in preventing this degradation. The most effective stabilizing buffer over all storage conditions was the UAS, enabling adequate recovery of the T790M variant by ddPCR. From  a technical point of view, stabilizing buffers and adequate storage conditions are a prerequisite for translation of urinary cfDNA diagnostics into clinical routine.

A weakness and well discussed next step for this group would be   that these experiments  should be confirmed in patients (harbouring tumors with mutations)  as the study is based on artificial spike in of DNA. 

  Overall, the study  should be published as it offers important information about stabilizers (pre-analytical considerations)   that may allow labs to use urine for cancer detection and other applications.

Author Response

Reviewer 2:

A weakness and well-discussed next step for this group would be that these experiments should be confirmed in patients (harbouring tumors with mutations) as the study is based on artificial spike in of DNA. Overall, the study should be published as it offers important information about stabilizers (pre-analytical considerations) that may allow labs to use urine for cancer detection and other applications.

We thank the reviewer for this positive feedback. Further studies using samples from patients with breast cancer are planned by our group.

Round 2

Reviewer 1 Report

Comments and Suggestions for Authors

Dear authors

Thank you for the additional comments and clarifications. Although central question are not solved the paper gives valuable insights in the problematic and describes the model system for further in depth studies accurately.